# Vitamin D Supply of Twins during Fetal Life, Its Relation to Anthropometric Parameters of Newborns and the Analysis of Other Factors Related to Birth Size

**DOI:** 10.3390/nu16203535

**Published:** 2024-10-18

**Authors:** Regina Ewa Wierzejska, Barbara Wojda, Dorota Agata Bomba-Opoń, Iga Rzucidło-Szymańska, Robert Brawura-Biskupski-Samaha, Iwona Szymusik

**Affiliations:** 1Department of Nutrition and Nutritional Value of Food, National Institute of Public Health NIH–National Research Institute, Chocimska St. 24, 00-791 Warsaw, Poland; bwojda@pzh.gov.pl; 2Department of Gynecology and Obstetrics, Institute of Medicine Collegium Medicum, Jan Kochanowski University, 25-317 Kielce, Poland; 3Department of Obstetrics and Perinatology, National Medical Institute of the Ministry of the Interior and Administration, 02-507 Warsaw, Poland; 4Department of Obstetrics, Perinatology and Neonatology, Centre of Postgraduate Medical Education, Cegłowska, 01-809 Warsaw, Poland

**Keywords:** vitamin D concentration, twin pregnancy, umbilical cord blood, neonatal parameters

## Abstract

Background/Objectives: Vitamin D deficiencies are very common in pregnant women, raising concerns about adverse health outcomes in children. This issue has hardly been studied in multiple pregnancies, the prevalence of which has been steadily increasing. Therefore, our study investigated the relationship between newborns’ anthropometric parameters and the concentration of 25(OH)D in maternal blood of women with twin pregnancies and umbilical cord blood. Methods: The study included 50 women who gave birth after the 36th week of twin gestation. The concentration of 25(OH)D was determined in maternal blood collected during the antenatal period and in the umbilical cord blood of 100 newborns. Anthropometric parameters of the newborns (birth weight, length and head and chest circumference) were obtained from hospital records. Data on nutrition and lifestyle during pregnancy were collected from the patients during an interview conducted by a dietitian. Results: No relationship between maternal and neonatal cord blood vitamin D concentrations and any of the anthropometric parameters of the newborns was found. However, only 6% of the mothers and 13% of the newborns had vitamin D deficiency (≤20 ng/mL). The type of pregnancy and maternal height were the main factors associated with neonatal size. Newborns from dichorionic pregnancies were on average 202 g heavier (*p* < 0.001) and 1 cm longer (*p* = 0.006) than newborns from monochorionic pregnancies. Newborns of mothers ≤160 cm in height had on average 206 g lower birth weight (*p* = 0.006) and were 3.5 cm shorter (*p* = 0.003) compared to newborns of taller mothers. Conclusions: Therefore, in our study, the neonatal size of twins was not related to the vitamin D status but to other factors such as the type of pregnancy and maternal height.

## 1. Introduction

Pregnant women are at high risk of vitamin D deficiency. It is estimated that it affects 46–98% of women worldwide [1,2,3]. There is currently very little separate data on women with twin pregnancies, which does not allow to draw general conclusions. However, the experts indicate that hypovitaminosis D may be even more pronounced in this population [4,5]. The concentration of vitamin D metabolite calcidiol [25(OH)D] in the umbilical cord blood of newborns is strongly correlated with maternal levels. According to estimates, it can reach an average of 80% of the maternal value [6,7,8], although some studies have shown that the concentration in umbilical cord blood is higher than in the mothers [9,10,11]. The concentration of the active metabolite of vitamin D calcitriol [1,25(OH)_2_D] has not been fully elucidated, although it should be noted that the placenta is able to convert calcidiol to calcitriol due to the production of 1 α-hydroxylase [12,13].

The most well-known and documented role of vitamin D during pregnancy is its participation in fetal skeletal growth and mineralization [6,12]. The knowledge on the impact of this nutrient on pregnancy outcomes is much more limited. Some studies indicate that low vitamin D concentrations increase the risk of preeclampsia, gestational diabetes, premature birth and affect newborn size [14,15,16,17]. However, some authors have expressed doubts [18] or even denied [19] this relationship.

In light of the scientific literature, the reference value of vitamin D concentration in pregnant women is an equally debatable issue. Some experts believe that the normal 25(OH)D concentration during pregnancy is above 30 ng/mL [20,21,22,23], others suggest >20 ng/mL [24,25,26,27] while some researchers diagnose vitamin D deficiency only with concentrations below 12 ng/mL [28]. There is also no consensus regarding the concentration of vitamin D in the umbilical cord blood of the newborn. A deficit is sometimes considered to be <20 ng/mL [29,30] or <12 ng/mL [31], while some experts believe that the normal level of vitamin D in umbilical cord blood is ≥30 ng/mL [23]. Such significant discrepancies have so far prevented a discussion on whether the criteria for saturation of the body with vitamin D and the guidelines for its supplementation should be the same for women in multiple and in singleton pregnancies.

So far, studies on the impact of vitamin D concentrations during pregnancy on neonatal anthropometric parameters have been conducted almost exclusively among women in singleton pregnancies. Our study included women with twin pregnancies and investigated whether vitamin D concentrations in maternal and cord blood (expressed as 25(OH)D) were associated with neonatal birthweight, length and head and chest circumference.

## 2. Materials and Methods

### 2.1. Study Design

The study was conducted among 50 women with twin pregnancies and their 100 newborns delivered after 36 weeks of gestation at the 1st Department of Obstetrics and Gynaecology, Medical University of Warsaw (2021–2022) and at the Department of Obstetrics, Perinatology and Neonatology, Centre of Postgraduate Medical Education in Warsaw (2023). Although delivery before 37 completed weeks of gestation is considered preterm, it is recommended to deliver uncomplicated monochorionic twins between 36 and 37 weeks due to the lowest rates of neonatal complications [32]. The exclusion criteria for patients were age below 18 years, non-Polish nationality and chronic liver and kidney diseases likely to affect vitamin D metabolism (liver cirrhosis, chronic hepatitis with elevated liver enzymes and chronic kidney failure). In addition, pregnancies complicated by twin-to-twin transfusion syndrome or with congenital anomalies of either of the twins were also excluded. Upon obtaining the informed consents to participate in the study, venous blood was collected from the patients during the antenatal period and umbilical cord blood at the delivery, thereby creating “mother–newborns” blood sets. The study was approved by the Bioethics Committee of the National Institute of Public Health, National Institute of Hygiene–National Research Institute in Warsaw under No. 6/2021.

### 2.2. Laboratory Analysis and Data Collection

The blood samples collected from the study participants were tested for total 25(OH)D using immunological tests (LIAISON^®^ 25 OH Vitamin D TOTAL Assay; DiaSorin Inc., Stillwater, MN 55082, USA). The lower detection threshold for vitamin D is 4.0 ng/mL. The intraassay and interassay CV were <8% and <10%, respectively. A 25(OH)D concentration of ≥30–50 ng/mL was considered optimal, >20–30 ng/mL as suboptimal, ≤20 ng/mL as deficiency and >50 ng/mL was considered high [20,21,22]. In cord blood, ≥20 ng/mL was the recommended level and <20 ng/mL indicated vitamin D deficiency [29,30].

Data on newborns (gender, birth weight, length, head and chest circumference and Apgar score at 5 min) were obtained from the hospital’s medical records. Anthropometric measurements were taken by midwives immediately after birth. Birth weight was measured using a beam scale. Measurement of the newborn’s length and head and chest circumference was performed according to generally applicable standards.

Birth weight discordance was calculated using the formula: birth weight of larger twin–birth weight of smaller twin/birth weight of the larger ×100. According to the guidelines of the American College of Obstetricians and Gynecologists, a cut-off of ≥20% was used to define significant weight discordance [33].

The patients’ intake of vitamin D (from fish, eggs, margarine (mandatorily fortified with vitamin D in Poland), butter, milk and dairy products), calcium (from milk and dairy products) and caffeine (from coffee and tea) was estimated during an interview conducted by a dietitian, based on the validated Food Frequency Questionnaire. Patients were also asked to list any vitamin/mineral preparations they were taking and for how long, as well as to provide information on the course of their pregnancy, body weight before conception, weight gain during pregnancy, smoking habits and sociodemographic data. Gestational weight gain was assessed based on the American Institute of Medicine guidelines for women in twin pregnancies [34]. Given that our study included pregnant women whose gestation exceeded 36 weeks, it is worth emphasizing that the assessment of their gestational weight gain is in line with the above-mentioned guidelines, which were developed on the basis of the weight gain of women who gave birth after the end of the 36th week of pregnancy to twins weighing not less than 2500 g. The minimum gestational length of 36 weeks for assessing the weight gain in women in twin pregnancies in the context of the guidelines is also used by other authors [35]. Patients’ characteristics are presented in Table 1.

### 2.3. Statistical Analysis

The following descriptive statistics were determined for the variables: percentage frequencies for qualitative variables, arithmetic mean with standard deviation for quantitative variables with a normal distribution and median and range of variability for other quantitative variables. Normality of distribution was tested using the Kolmogorov–Smirnov test. In particular, vitamin D concentrations in maternal and cord blood as well as neonatal birth weight were subject to normal distribution. The distribution of neonatal body length and head/chest circumference did not correspond to a normal distribution.

Simple linear regression was used to test whether there was a relationship between maternal and umbilical cord blood vitamin D concentrations and the newborn’s body weight, length, head circumference and chest circumference. It was assessed whether a change in vitamin D concentration corresponds to a statistically significant change in the value of an anthropometric parameter. We report the corresponding regression coefficient (slope) with the standard error (SE), the 95% confidence interval (CI) limits and the statistical significance of its difference from zero (*p*-value).

Linear regression was also used to assess the relationship between maternal and child blood vitamin D concentrations. In this case, the Pearson correlation coefficient was also calculated. In comparisons between two categories distinguished on the basis of a qualitative variable, the Student’s *t*-test was applied in neonatal weight analyses (the data met its assumptions). The non-parametric Mann–Whitney test was used for the other anthropometric data.

The significance level of 0.05 was adopted for all statistical analyses. They were carried out using SPSS software version 12.0 PL.

## 3. Results

### 3.1. Vitamin D Status in Mothers and Newborns

Vitamin D deficiency (≤20 ng/mL) was found in 6% of women and 13% of newborns (Figure 1). Although the newborns had markedly lower concentrations of 25(OH)D than the mothers (24.4 ± 5.9 vs. 39.7 ± 10.7 ng/mL), there was a strong correlation between the level of 25(OH)D in maternal and cord blood (the Pearson linear correlation coefficient is 0.653; *p* < 0.001). In the linear regression model, an increase in maternal 25(OH)D concentration by one unit corresponds to an increase in newborn’s level by 0.34 (SE = 0.04) units (*p* < 0.001) (Table 2).

The percentage of vitamin D deficient newborns decreased with the higher vitamin D blood saturation in the mothers. All neonates born to women with deficiency were also vitamin D deficient. Among women with suboptimal vitamin D concentration, 29% gave birth to children with a deficiency in comparison to 9% of women with optimal levels. None of the neonates born to mothers with high vitamin D concentrations had a deficit of this nutrient. Vitamin D supplements were taken by 98% of the mothers, with the median dose of 2000 IU (50 µg).

### 3.2. Neonatal Weight

No statistically significant association was found between the concentration of 25(OH)D both in maternal and cord blood and neonatal birth weight (Table 2), despite a wide range of concentrations being observed: 15.1–64.4 ng/mL in mothers and 10.6–39.5 ng/mL in newborns. The character of the relationship between, for example, the birth weight of the newborn and the mother’s vitamin D concentration is shown in Figure 2. There was also no statistically significant difference in the birth weight of children with diagnosed deficiency of vitamin D in umbilical cord blood and those with the normal concentrations (2642 g vs. 2545 g).

Neonatal birth weight was related to several other analyzed factors i.e., pregnancy type, gestational weight gain and maternal height. The average birth weight of neonates from dichorionic pregnancies was 202 g higher compared to neonates from monochorionic pregnancies (2651 g vs. 2449 g) (*p* < 0.001). The average birth weight of neonates born to mothers with excessive weight gain was 151 g higher compared to children born to mothers with normal weight gain (2727 g vs. 2576 g) (*p* = 0.035). The minimum and maximum birth weight of newborns delivered by women with excessive weight gain was 2350 g and 3030 g, respectively, compared to 1935 g and 3270 g in children born to women with normal weight gain. The average birth weight of children born to mothers ≤160 cm was 206 g lower than children born to taller mothers (2368 g vs. 2574 g) (*p* = 0.006). Also, all neonates born to short-statured women had low birth weight (<2500 g). The birth weight differences among newborns of mothers with and without hypertension during pregnancy were of borderline statistical significance (*p* = 0.071) (2346 g vs. 2571 g). The other analyzed factors (maternal age, education, place of residence, number of pregnancies, pregestational BMI, season of delivery, vitamin D intake from diet and supplements, calcium intake, caffeine intake, use of multivitamin preparations, gestational diabetes, anemia and sex of the newborn) had no statistically significant relationship with neonatal birth weight.

### 3.3. Neonatal Length

Similarly to body weight, neonatal length had no statistically significant relationship with 25(OH)D concentration in maternal and cord blood (Table 2). There were also no significant differences in body lengths among newborns with and without vitamin D deficiency. Statistically significant factors included only maternal height, the type of pregnancy and the women’s age. The average (median) length of the neonates from dichorionic twin pregnancies was 1 cm greater compared to the neonates from monochorionic pregnancies (51 cm vs. 50 cm) (*p* = 0.006). Children born to mothers ≤160 cm tall were on average (median) 3.5 cm shorter compared to children of taller mothers (47.5 cm vs. 51 cm) (*p* = 0.003). Children of mothers aged ≥27 years were on average (median) 1.5 cm longer than children of younger mothers (*p* = 0.038). As in the case of neonatal birth weight, the length difference in the neonates born to women with and without gestational hypertension was of borderline statistical significance (*p* = 0.072) (49.5 cm vs. 51 cm).

### 3.4. Head Circumference

The distribution, but not the average value of head circumference was statistically significantly related to pregestational body weight of the mother and the sex of the neonate. Although the average head circumference of infants born to mothers with normal and excessive body weigh was the same (median 33 cm), the distribution of results was different. The most prevalent head circumference (dominant value) in children born to mothers with normal pregestational BMI was 33 cm vs. 34 cm in children born to mothers with excessive BMI (*p* = 0.012). The same was observed for the sex of the newborns. The average head circumference was the same for boys and girls (median 33 cm), but the range of head circumference variability was greater in boys (30–35 cm) compared to girls (31–34 cm) (*p* = 0.020).

### 3.5. Neonatal Chest Circumference

Neonatal chest circumference was statistically significantly related to several factors including pregnancy type, maternal height, gestational weight gain, gestational hypertension and sex of the neonate. Infants from dichorionic twin pregnancies had on average (median) 1 cm larger chest circumference compared to infants from monochorionic pregnancies (31 cm vs. 30 cm) (*p* < 0.05). In infants of women ≤160 cm tall, this parameter was on average (median) 2 cm lower than in infants of women over 160 cm (29 cm vs. 31 cm) (*p* = 0.005). The chest circumference of infants born to mothers with excessive weight gain was on average (median) 1 cm larger than that of infants born to mothers with normal weight gain (32 cm vs. 31 cm) (*p* = 0.007). Children born to mothers with hypertension had on average (median) 2 cm smaller chest circumference than children of mothers without hypertension (29 cm vs. 31 cm) (*p* = 0.046), and boys had on average (median) 1 cm larger chest circumference than girls (31 cm vs. 30 cm) (*p* = 0.034).

## 4. Discussion

To the best of our knowledge, the relationship between vitamin D status in women with twin pregnancies and the birth weight of their newborns has been investigated only in two published studies conducted in China [36] and Italy [37]. Both studies found no association between maternal 25(OH)D concentrations and neonatal birth weight, although in the first study almost 80% of children had vitamin D deficiency at birth, whereas in the second study 38% of mothers had vitamin D levels below 30 ng/mL in the third trimester of pregnancy. Our study also did not show any relationship between maternal or cord blood 25(OH)D levels and any of the anthropometric parameters of the newborns routinely measured at delivery, although with the same deficiency criterion as in the Chinese study (vitamin D concentration <20 ng/mL), only 13% of our newborns had vitamin D concentration below guideline values. The significant difference in the percentage of deficient newborns is probably related to the fact that Polish women generally take much higher doses of vitamin D than women in China (2000 IU vs. 500 IU).

In light of the recommendations of various experts in Poland at the time of our study, pregnant women should take 2000 IU of vitamin D daily according to some experts [20,38] or 1500–2000 IU according to others [39]. Thus, the results of the study would indicate that the majority of women adhere to these recommendations, which is probably due to the fact that the composition of vitamin/mineral preparations for pregnant women has improved in recent years [40]. However, it is worth mentioning that the World Health Organization considers that vitamin D supplementation should apply only to women with suspected deficiency, and in such a situation the dose of 200 IU per day is sufficient [41]. Globally, this results in different doses of vitamin D being used even in clinical trials, which, in the absence of an assessment of its concentration at the beginning of pregnancy, means that the effects of supplementation on the course of pregnancy and the health of the child are inconclusive [2,42].

Numerous studies have been conducted among women in singleton pregnancies. In an observational study in the UK, no differences in neonatal body parameters were found between groups of mothers with vitamin D levels <12 ng/mL vs. >30 ng/mL [43]. In an Australian study, maternal vitamin D deficiency (<20 ng/mL) did not correlate with lower neonatal weight and length, but such low levels were only found in <7% of the mothers, probably due to the widespread supplementation of this nutrient (declared by 98% of the patients) [1]. No association between maternal and cord blood vitamin D concentrations and any of the neonatal anthropometric parameter [44] or fetal femoral length [45] was found in Polish studies, despite the fact that 50% of the mothers and almost 30% of the newborns had vitamin D concentrations below 20 ng/mL. On the other hand, a Chinese and a Turkish study found a correlation between neonatal parameters and vitamin D status of the mother. In the former study, infants born to deficient women had a slightly (by 65 g) but significantly lower birth weight than newborns of women with normal vitamin D status [46]. In the latter study, infants born to women supplementing vitamin D during pregnancy (58% of the group) were longer and had greater head and chest circumferences than infants born to women who did not use vitamin D supplementation [47]. Contradictory results regarding the relationship between maternal and umbilical cord blood vitamin D concentrations and anthropometric parameters of the newborns were obtained in a study conducted in Finland [48].

The results of a randomized, placebo-controlled trials (RCTs) which have so far only been conducted in women in singleton pregnancies are equally inconsistent. The above can be illustrated by several Iranian studies. Vaziri et al. [49] and Mohammad-Alizadeh-Charandabi et al. [50] showed that maternal vitamin D supplementation was not associated with newborns’ size, whereas Naghshineh and Sheikhaliyan [51] concluded that infants born to mothers taking vitamin D during pregnancy were heavier by an average of 230 g. In an Indian study, vitamin D supplementation in dosages depending upon blood concentrations was found to increase neonatal birth weight by an average of 200 g [52]. Roth et al. stated that infants born to mothers from the supplementation group did not differ in size from the newborns of mothers from the control group [53]. Finally, the conclusions from meta-analyses of RCTs are also inconsistent [2,54,55,56,57].

In our study, the anthropometric parameters of the newborns were associated with the type of pregnancy. Larger neonates were born to women in dichorionic pregnancies compared to monochorionic pregnancies, which is consistent with the current state of knowledge. In a large Dutch study, the mean birth weight of newborns from dichorionic pregnancies was 221 g greater compared to newborns from monochorionic pregnancies [58], which is very similar to the results of our study (202 g difference). It is worth adding that the median length of dichorionic pregnancy in both studies was one week longer than for monochorionic pregnancy (37 vs. 36 weeks). Higher birth weights of infants from dichorionic pregnancies compared to monochorionic pregnancies were also reported in Finland [59] and Thailand [60].

In twin pregnancies, as in singleton pregnancies, the growth of the fetus and the birth weight of the newborns are associated with gestational weight gain [35,44,61,62], which was also proven in our study. Children born to mothers with excessive weight gain were heavier than infants born to mothers with normal weight gain. It is worth noting that in twin pregnancies, which are associated with a higher risk of low birth weight, greater gestational weight gain may in some cases even promote better fetal growth of the newborns. In a large American study by Bodnar et al., the risk of delivering small-for-gestational-age neonates decreased with increasing gestational weight gain in women in twin pregnancies, while the risk of large-for-gestational-age neonates was observed to increase [61]. The same conclusion emerges from a meta-analysis of observational studies conducted among women in singleton pregnancy [62].

The relationship between maternal height and neonatal size determined in our study is also not surprising, as it has been quite well documented in women in singleton pregnancies. Maternal and paternal height reflects the genetic growth potential of the fetus, with taller mothers generally giving birth to larger children [63,64,65]. In addition to genetic factors, anatomical factors may also play an important role. Such factors can include a smaller pelvis or smaller uterus size in short-statured women, which can affect uterine expansion during pregnancy, the growth of the placenta and the fetus [65]. In our study, neonates born to mothers up to 160 cm tall were over 200 g lighter and markedly shorter (3.5 cm on average) compared to neonates born to mothers of taller stature (above 160 cm). In the Finnish study, birth weight, length and head circumference increased with maternal height, regardless of the sex of the neonate. For example, boys born to mothers ≤ 158 cm tall weighed on average 265 g less and were 1.3 cm shorter than boys born to mothers ≥ 173 cm tall [64]. These neonatal anthropometric parameters were also found to correlate with maternal height in an Austrian study [63], while in the American study, the mother’s height correlated with the birth weight of the newborns [66]. In the light of a meta-analysis of studies conducted in singleton pregnancies, short-statured women have a greater risk of giving birth to newborns with low birth weight [65].

As to the sex of the neonate, it is often related to birth size. Girls are often born smaller than boys [63,64], but there are also studies that do not show such a difference [39]. The newborn girls included in our study had smaller head and chest circumferences.

When interpreting our results, the limitations of the study should be taken into account. Firstly, the measurements of maternal 25(OH)D concentrations were performed in the perinatal period, which does not mean that they were typical throughout or during most of the pregnancy course. Therefore, it is not known whether newborns had a relatively constant supply of vitamin D in utero. Another limitation of our study is the relatively small sample size mostly due to the small overall population of women pregnant with twins. It is also worth noting that the study was carried out in two separate tertiary centers in Warsaw, which also run separate outpatient clinics for multiple gestations. It can therefore be assumed that women had better access to antenatal education, including guidelines on vitamin supplementation. Therefore, the study group may not adequately reflect vitamin D status of patients from smaller centers.

## 5. Conclusions

Our study did not demonstrate any relationship between neonatal anthropometric parameters, maternal 25(OH)D concentrations measured antenatally and cord blood concentrations. The main factors associated with neonatal size were the type of pregnancy (dichorionic vs. monochorionic), maternal height and the gestational weight gain.

## Figures and Tables

**Figure 1 nutrients-16-03535-f001:**
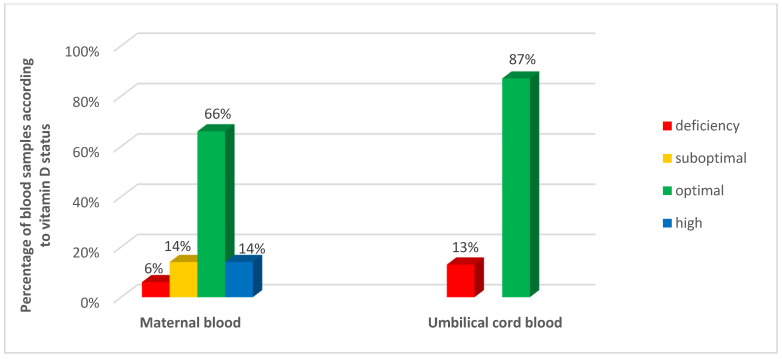
Vitamin D status in maternal and umbilical cord blood.

**Figure 2 nutrients-16-03535-f002:**
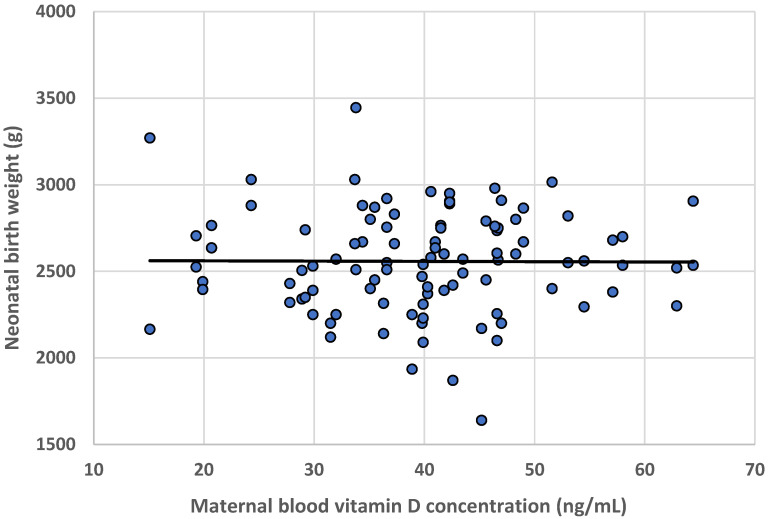
Vitamin D concentration in maternal blood vs. neonatal birth weight.

**Table 1 nutrients-16-03535-t001:** Maternal and neonatal characteristics.

Characteristic and Its Measures	Values in Studied Group
Number of women, n including: monochorionic pregnancies, n (%) * dichorionic pregnancies, n (%)	50 23 (46%) 27 (54%)
Age (in years), mean ± SD **	31.7 ± 4.6
Education, n (%) higher other	36 (72) 14 (28)
Place of residence, n (%) city/town rural/village	46 (92) 4 (8)
Number of pregnancies, n (%) first subsequent	26 (52) 24 (48)
Gestational age (in weeks), median (min-max) ***	36 (36–38)
Gestational age of monochorionic pregnancy (in weeks), median (min-max)	36 (36–37)
Gestational age of dichorionic pregnancy (in weeks), median (min-max)	37 (36–38)
Due date season, n (%) spring–summer autumn–winter	25 (50) 25 (50)
Maternal Body Mass Index (BMI) prior to conception, median (min-max)	22.5 (16.6–38.9)
Gestational weight gain, n (%) low normal excessive	24 (48) 20 (40) 6 (12)
Gestational diabetes, n (%)	10 (20)
Hypertension, n (%)	3 (6)
Anaemia, n (%)	17 (34)
Smoking during pregnancy, n (%)	0 (0)
Vitamin D supplementation, n (%)	49 (98)
Daily vitamin D intake with food (µg), median (min–max) with food and dietary supplements (µg), median (min–max)	2.4 (0.4–9.1) 52.1 (1.2–154.6)
Supplementation with vitamin-mineral preparations (multicomponent), *n* (%)	49 (98)
Calcium intake from milk and dairy products (mg), median (min–max)	641.1 (0.0–2900.4)
Caffeine intake from coffee and tea (mg), mean ± SD	91.7 ± 73.2
Fish consumption at least once a week, n (%)	20 (40)
Sex of the newborn, n (%) male female	45 (45) 55 (55)
Neonatal weight (g), mean ± SD	2557.8 ± 295.6
Number of newborns with low birth weight (<2500 g), n (%)	40 (40)
Number of twin pairs with low birth weight (<2500 g), n (%)	12 (24)
Birth weight discordance, n (%)	6 (12%)
Neonatal length (cm), median (min–max)	51 (46–56)
Neonatal head circumference (cm), median (min–max)	33 (30–35)
Neonatal chest circumference (cm), median (min–max)	31 (27–35)
Apgar score at 5 min (points), median (min–max)	10 (8–10)
Maternal 25(OH)D concentration (ng/mL), mean ± SD	39.7 ± 10.7
Maternal 25(OH)D concentration in women in monochorionic pregnancies (ng/mL), mean ± SD	39.0 ± 9.4
Maternal 25(OH)D concentration in women in dichorionic pregnancies (ng/mL), mean ± SD	40.3 ± 11.6
Cord blood 25(OH)D concentration (ng/mL), mean ± SD	24.4 ± 5.9
Cord blood 25(OH)D concentration in neonates born from monochorionic pregnancies (ng/mL), mean ± SD	24.5 ± 5.2
Cord blood 25(OH)D concentration in neonates born from dichorionic pregnancies (ng/mL), mean ± SD	24.4 ± 6.4
Cord blood 25(OH)D concentration in first-born twins, (ng/mL), mean ± SD	24.8 ± 6.0
Cord blood 25(OH)D concentration in second-born twins, (ng/mL), mean ± SD	24.1 ± 5.8

* for qualitative data. ** for quantitative data with a normal distribution. *** for quantitative data with other distribution.

**Table 2 nutrients-16-03535-t002:** Association quantitative characteristics of newborn with maternal and umbilical cord blood vitamin D concentrations; results of simple linear regression.

Outcome Variable Concerning Newborn	Regression Coeficient (Slope)	SE	95% CI	*p*-Value
Independent variable: vitamin D concentration in maternal blood				
Concentration vit. D in cord blood (ng/mL)	0.34	0.04	0.25–0.42	<0.001
Weight (g)	−0.16	2.79	−5.71–5.38	NS *
Length (cm)	0.01	0.02	−0.03–0.05	NS *
Head circumference (cm)	−0.002	0.011	−0.024–0.019	NS *
Chest circumference (cm)	0.01	0.01	−0.02–0.04	NS *
Independent variable: vitamin D concentration in umbilical cord blood				
Weight (g)	−9.1	5.0	−19.0–0.07	NS *
Length (cm)	−0.05	0.04	−0.12–0.02	NS *
Head circumference (cm)	−0.03	0.02	−0.06–0.01	NS *
Chest circumference (cm)	−0.03	0.02	−0.08–0.02	NS *

* not significant statistically.

## Data Availability

The raw data supporting the conclusions of this article will be made available by the authors, without undue reservation.

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
