# Peer review of "Vitamin D Supply of Twins during Fetal Life, Its Relation to Anthropometric Parameters of Newborns and the Analysis of Other Factors Related to Birth Size"

_nutrients, 2024, doi:10.3390/nu16203535_

Round 1

Reviewer 1 Report

Comments and Suggestions for Authors

The authors of the manuscript “Vitamin D supply of twins during fetal life, its relation to anthropometric parameters of newborns and the analysis of other factors related to birth size” demonstrated that neonatal size of the twins was not related to the vitamin D status but to other factors such as the type of pregnancy and maternal height. Some questions arise while reading the paper:

In the methods, in the exclusion criteria section, what is meant by chronic renal and liver disease? Please specify better

Please specify how the sample size was calculated

Are there any differences regarding the parameters of newborns between mothers who had a vitamin D lower than 20 ng/ml and those with higher values?

Author Response

The authors of the manuscript “Vitamin D supply of twins during fetal life, its relation to anthropometric parameters of newborns and the analysis of other factors related to birth size” demonstrated that neonatal size of the twins was not related to the vitamin D status but to other factors such as the type of pregnancy and maternal height.

Some questions arise while reading the paper: In the methods, in the exclusion criteria section, what is meant by chronic renal and liver disease? Please specify better

The relevant information has been added.

Please specify how the sample size was calculated

The sample size was not calculated before the study began. The study was planned for 2021-2023. Taking into account the inclusion and exclusion criteria, the required patient consents, and the fact that the delivery was planned and occurred during the day (dealing with biological material and consents) finally we could conduct our study on 50 patients.

We must also mention that the COVID-19 pandemic broke out right after the study began and the recruitment in 2021 was greatly limited. The study was conducted in two separate tertiary centers in Warsaw: 1 st Department of Obstetrics and Gynaecology, Medical University of Warsaw (years 2021-2022) and the Department of Obstetrics, Perinatology and Neonatology, Centre of Postgraduate Medical Education (2023). It was due to the fact that two researchers (IS and RBBS) who dealt with twin pregnancies changed place of work from the first hospital to the second by the end of 2022.

We hope that these explanations are satisfactory.

Are there any differences regarding the parameters of newborns between mothers who had a vitamin D lower than 20 ng/ml and those with higher values?

Only 3 mothers had vitamin D deficiency. The difference between children born to mothers with deficiency and children of mothers without deficiency (6 vs. 94) concerned only the concentration of vitamin D in umbilical cord blood. The anthropometric parameters of the children did not differ significantly (see the table below). We did not insert this table into the manuscript due to the small number of mothers with deficiency, but if you think it should be inserted, we can do it.

Newborn characteristic and its measures

Maternal vit. D ≤ 20 ng/mL

Maternal vit. D >20 ng/mL

Statistical test; p-value

Cord blood 25(OH)D concentration (ng/mL), mean ± SD

11.5 ± 0.9

25.3 ± 5.1

Student t – test; p<0,001

Neonatal weight (g), mean ± SD

2583 ±347

2556 ±292

Student t – test; NS

Neonatal length (cm), median (min-max)

52 (49-56)

51 (46-56)

Mann-Whitney test; NS

Neonatal head circumference (cm), median (min-max)

33.25 (32-34)

33 (30-35)

Mann-Whitney test; NS

Neonatal chest circumference (cm), median (min-max)

30.5 (29-33)

31 (27-35)

Mann-Whitney test; NS

Reviewer 2 Report

Comments and Suggestions for Authors

The manuscript titled 'Vitamin D supply of twins during fetal life, its relation to anthropometric parameters of newborns and the analysis of other factors related to birth size' is well written and researched. However, it needs additional details and tables in the methods and results sections to make it shine.The methods section needs to provide relevant information about all parameters and where they follow the normal distribution or not.  Descriptive tables should also include p-values for clarity. The statistical program used (SPSS version 12) is dated, as it from 2003, and a more recent version would be better if possible. All relevant information on the linear model methodology as well as the parameters the model was corrected with should be fully stated here. The results section should be expanded to include more detailed tables showing all statistical tests and comparisons, particularly for vitamin D and neonatal measurements. Each table should clearly state the statistical test used and the associated p-values. It is advisable to follow an example from the Journal on how to format such tables as the descriptives table lacks headings. Figure 1 needs labelled axes, particularly clarification of what the percentage on the 'Blood samples' axis represents, and the x-axis in Figure 2 should start at zero. A table summarizing the linear regression analysis is also required, and the methods section should include all relevant regression metrics for quality checking as well.

The discussion is generally sound and well explained, but would benefit from a small paragraph on national vitamin D policies and their potential influence on the results of the study and/or comparison with the other relevant studies . It would also be valuable to include the authors' opinions on potential vitamin D levels and  on how these national policies may act as a confounding factor in vitamin D research, especially when comparing national averages.

Author Response

The manuscript titled 'Vitamin D supply of twins during fetal life, its relation to anthropometric parameters of newborns and the analysis of other factors related to birth size' is well written and researched.

However, it needs additional details and tables in the methods and results sections to make it shine.

The methods section needs to provide relevant information about all parameters and where they follow the normal distribution or not.  Descriptive tables should also include p-values for clarity. The statistical program used (SPSS version 12) is dated, as it from 2003, and a more recent version would be better if possible.

In our study only basic statistical analysis methods (descriptive statistics, most popular statistical tests, linear regression) were used, which are also implemented in SPSS 12.PL and this package was used. We do not have a license for newer versions of the program.

All relevant information on the linear model methodology as well as the parameters the model was corrected with should be fully stated here. 

The relevant information in the Statistical Analysis section has been rewritten

The results section should be expanded to include more detailed tables showing all statistical tests and comparisons, particularly for vitamin D and neonatal measurements. Each table should clearly state the statistical test used and the associated p-values. It is advisable to follow an example from the Journal on how to format such tables as the descriptives table lacks headings.

According to the reviewer's suggestion, we added a table with linear regression results confirming the lack of correlation between vitamin D concentration in the mother and newborn and anthropometric parameters. In addition, we specify the type of statistical tests in more detail in the Statistical Analysis section. P-values ​​for statistically significant results are given, and for insignificant ones they are defined as NS.

The heading in Table 1 has been added.

Figure 1 needs labelled axes, particularly clarification of what the percentage on the 'Blood samples' axis represents, and the x-axis in Figure 2 should start at zero.

The vertical axis has been improved. At the same time we would like to clarify that Figure 2 does not start from the point (0,0) because there were no such low vitamin D levels or neonates with a body weight below 1500 g.

In our opinion, the new version of the figure (incloused in PDF) looks worse than the previous one, but we can change it if you think it is necessary.

 A table summarizing the linear regression analysis is also required, and the methods section should include all relevant regression metrics for quality checking as well.

The relevant information has been added.

The discussion is generally sound and well explained, but would benefit from a small paragraph on national vitamin D policies and their potential influence on the results of the study and/or comparison with the other relevant studies . It would also be valuable to include the authors' opinions on potential vitamin D levels and  on how these national policies may act as a confounding factor in vitamin D research, especially when comparing national averages.

Thank you. This data has been supplemented.

Round 2

Reviewer 2 Report

Comments and Suggestions for Authors

I am pleased with the answers and I think its ok to proceed to publication